# Inhibitors of 3-Hydroxy-3-methylglutaryl Coenzyme A Reductase Decrease the Growth, Ergosterol Synthesis and Generation of *petite* Mutants in *Candida glabrata* and *Candida albicans*

**DOI:** 10.3390/ijms242316868

**Published:** 2023-11-28

**Authors:** Dulce Andrade-Pavón, Eugenia Sánchez-Sandoval, Joaquín Tamariz, Jose Antonio Ibarra, César Hernández-Rodríguez, Lourdes Villa-Tanaca

**Affiliations:** 1Laboratorio de Biología Molecular de Bacterias y Levaduras, Departamento de Microbiología, Escuela Nacional de Ciencias Biológicas, Instituto Politécnico Nacional, Prolongación de Carpio y Plan de Ayala S/N, Col. Santo Tomás, Alc. Miguel Hidalgo, Ciudad de México 11340, Mexico or dandradep@ipn.mx (D.A.-P.); chdez38@hotmail.com (C.H.-R.); 2Core Facilities Department, Gothenburg University, Medicinaregatan 9 A-B, 41390 Göteborg, Sweden; eugensanch@gmail.com; 3Departamento de Química Orgánica, Escuela Nacional de Ciencias Biológicas, Instituto Politécnico Nacional, Prolongación de Carpio y Plan de Ayala S/N, Col. Santo Tomás, Alc. Miguel Hidalgo, Ciudad de México 11340, Mexico; jtamarizm@ipn.mx; 4Laboratorio de Genética Microbiana, Departamento de Microbiología, Escuela Nacional de Ciencias Biológicas, Instituto Politécnico Nacional, Prolongación de Carpio y Plan de Ayala S/N, Col. Santo Tomás, Alc. Miguel Hidalgo, Ciudad de México 11340, Mexico; jaig19@gmail.com

**Keywords:** α-asarone analogues, fibrates, HMGR, *C. glabrata*, *C. albicans*, yeast growth, ergosterol synthesis, *petite* mutants, antifungals

## Abstract

*Candida glabrata* and *Candida albicans*, the most frequently isolated candidiasis species in the world, have developed mechanisms of resistance to treatment with azoles. Among the clinically used antifungal drugs are statins and other compounds that inhibit 3-hydroxy-3-methylglutaryl coenzyme A reductase (HMGR), resulting in decreased growth and ergosterol levels in yeasts. Ergosterol is a key element for the formation of the yeast cell membrane. However, statins often cause DNA damage to yeast cells, facilitating mutation and drug resistance. The aim of the current contribution was to synthesize seven series of compounds as inhibitors of the HMGR enzyme of *Candida* ssp., and to evaluate their effect on cellular growth, ergosterol synthesis and generation of *petite* mutants of *C. glabrata* and *C. albicans.* Compared to the reference drugs (fluconazole and simvastatin), some HMGR inhibitors caused lower growth and ergosterol synthesis in the yeast species and generated fewer *petite* mutants. Moreover, heterologous expression was achieved in *Pichia pastoris*, and compounds **1a**, **1b**, **6g** and **7a** inhibited the activity of recombinant CgHMGR and showed better binding energy values than for α-asarone and simvastatin. Thus, we believe these are good candidates for future antifungal drug development.

## 1. Introduction

Fungal infections are frequently caused by yeasts of the *Candida* genus, and the first and second most commonly isolated species in patients are *C. albicans* and *C. glabrata*, respectively [1,2,3]. Although the treatment of patients suffering from candidiasis with one of the clinical antifungal drugs can in many cases eliminate the yeast, problems are often encountered due to the increasing presence of drug-resistant strains [1,4,5,6]. Considering the number of cases with resistance to conventional antifungal agents, there is a need for new drugs capable of ensuring the efficacy of treatments and decreasing their toxicity.

Statins have recently been evaluated as inhibitors of 3-hydroxy-3-methylglutaryl CoA reductase (HMGR), an enzyme that catalyzes the conversion of HMG-CoA to mevalonate, which is used by the cell as a precursor for the production of sterols, coenzyme Q10 and dolichol [7,8,9]. Among the sterols is ergosterol, a key element for the formation of the yeast cell membrane. Additionally, statins have been shown to lower the concentration of intracellular esters by the competitive inhibition of the HMGR enzyme [10], thus affecting the viability of *Candida* yeasts. Exposure to statins leads to a 70% drop in ergosterol levels in *C. glabrata*, thereby altering the activity of membrane-bound enzymes as well as disrupting mitochondrial function and increasing membrane permeability [9,10,11].

Apart from their action as growth inhibitors, statins also have a synergistic effect with fluconazole and itraconazole. The antifungal activity is improved on *C. albicans*, *C. tropicalis*, *C. parapsilosis* and *Cryptococcus neoformans* when fluconazole is co-administered with a statin, even in resistant strains. Hence, this relationship would seem to be beneficial for the treatment of patients with chronic or recurrent candidiasis [12,13]. However, statins have been found to cause hepatotoxicity by inducing apoptosis and sometimes myopathy in liver tissue [14,15,16,17]. In *C. glabrata*, they are known to trigger the formation of *petite* mutants, which are yeast strains with mtDNA damage that can lead to drug resistance [9]. 

Given the side effects of statins in humans and yeasts, new drugs with better inhibitory activity on HMGR have been sought. Accordingly, numerous lipid-lowering compounds have been synthesized [18,19]. These compounds have been proven to inhibit the HMGR in *Schizosaccharomyces pombe* and *Homo sapiens* proportional to their concentration [18]. Since the catalytic sequence of HMGR is highly conserved in different species (as shown with the analysis of the amino acid sequence of this enzyme in various fungi), the same compounds could possibly inhibit growth in other yeasts [20].

Because the inhibition of HMGR results in a decrease in the synthesis of ergosterol, a key factor for the viability of the yeast cell membrane, this enzyme is suggested as a therapeutic target [21]. The aim of the current contribution was to evaluate the effect of the series of test compounds **1a**–**b**, **2a**–**c**, **3a**–**b**, **4a**–**c**, **5a**–**b**, **6a**–**g** and **7a**–**b** on the capacity of *C. albicans* and *C. glabrata* to grow, synthesize ergosterol and generate *petite* mutants. Moreover, the heterologous expression of HMGR of *C. glabrata* was achieved in *Pichia pastoris* to examine the inhibition of this enzyme by the test compounds.

## 2. Results

### 2.1. Schematic Representation and Prediction of the Secondary Structure of the HMGRs of C. glabrata and C. albicans

In *C. glabrata* and *C. albicans*, the sequences of the genes coding for the HMGR protein are herein denominated CgHMGR and CaHMGR. Although the open reading frame (ORF) encoding for each of the proteins analyzed varied in size, both shared characteristics of class I HMGRs, including a transmembrane region in the N-terminal, a soluble region in the C-terminal and a linker between these two regions. As portrayed in the diagrams, the soluble region of CgHMGR (Figure 1a) and CaHMGR (Figure 1b) consists of three domains: an N-domain that connects the L-domain to the linker domain. The L-domain contains an HMG-CoA binding region, while the function of the S-domain is to bind to NADPH. The figures illustrate the amino acids involved in the catalytic site: K (Lys), D (Asp), E (Glu) and H (His). CgHMGR and CaHMGR have eight and nine transmembrane segments, respectively, but are highly conserved in the soluble region. Consequently, the HMGR inhibitors currently examined could possibly have activity on both enzymes.

### 2.2. Determination of the MIC_50_ of Fluconazole, Simvastatin and the Test Compounds on Different Strains of C. glabrata and C. albicans, as well as Its Physicochemical Properties

The examination of the susceptibility of *Candida* strains to fluconazole, simvastatin and the test α-asarone-based compounds (**1a**–**b**, **2a**–**c**, **3a**–**b**, **4a**–**c**, **5a**–**b**, **6a**–**g** and **7a**–**b**) (Figure 2), carried out by means of the microdilution method, showed three phenotypes (S (sensitive), SDD (sensitive dose dependent) and R (resistant)). The 50% minimum inhibitory concentration (MIC_50_) ranged from 8 to 128 μg/mL for the three *C. glabrata* strains and 4 to 128 μg/mL for the three *C. albicans* strains (Table 1). By treating these strains with simvastatin, the MIC_50_ values required for the *C. glabrata* strains coincide with the phenotypes obtained with fluconazole (8, 32 and 128 μg/mL for S, SDD and R, respectively). The same order of MIC_50_ values was found for the strains of *C. albicans* (S, SDD and R, from lower to higher).

When the test compounds (designed as inhibitors of HMGR) were evaluated, the MIC_50_ values for **1a**, **1b** and **7a** were less than 256 μg/mL. Whereas compound **1a** only exhibited inhibitory activity on CBS138 and CAL18 and **1b** only on ATCC10231 and CAL18, **7a** was able to inhibit all six strains of *C. glabrata* and *C. albicans*, emphasizing the advantage of the latter compound. The MIC_50_ of the other compounds was greater than 256 for all six strains under study; this could be due to the physicochemical properties that are shared between them (Appendix A).

### 2.3. Effect of the Compounds on the Growth of C. glabrata and C. albicans

The effect of the compounds on the growth of *C. glabrata* (CBS138) and *C. albicans* (ATCC10231) was estimated by comparing the viable cell count of treated and control (untreated) yeasts. Growth kinetics was monitored at 3 and 12 h and conducted with various concentrations of the compounds. The test compounds (series **1**–**7**) showed growth inhibition (Appendix A), with a greater decrease in viability on *C. glabrata* than *C. albicans*, even at the lowest concentration (100 µM). The activity of the test compounds was similar to that of the reference drugs at 600 µM. 

After treating each yeast strain with one of the test compounds (at concentrations of 100, 300 and 600 µM) for 24 h, the serial dilutions made with the resulting growth were dripped onto YPD plates to establish the susceptibility profile for *C. glabrata* (Figure 3a) and *C. albicans* (Figure 3b). All compounds reduced yeast growth compared to untreated yeasts. Additionally, the treatment with the highest concentration of the compounds provided total inhibition of the growth of both species.

### 2.4. Effect of the Test Compounds on Ergosterol Synthesis in C. glabrata and C. albicans

Since the compounds are inhibitors of an enzyme involved in the synthesis of ergosterol, the amount of this sterol present in the treated and untreated yeasts was determined. Yeasts were treated with the test compounds and fluconazole. Absorption spectra characteristic of ergosterol were obtained, and the height of the peaks represented the level of this sterol. The percentage decrease in the level of ergosterol was calculated in relation to untreated cells (±the standard deviation derived from four samples from two independent experiments) with the equation described in Materials and Methods. The absorption spectra of *C. glabrata* and *C. albicans* treated with each of the compounds can be seen in Appendix A. The compounds with a significantly better inhibitory activity than that obtained with simvastatin are indicated with an asterisk (Table 2). For some compounds, the absence of a significant difference reflects an effect on the synthesis of ergosterol similar to that of the reference compound. According to these results, the compounds acted on the target for which they have been designed, and the majority of them showed better inhibition than simvastatin.

### 2.5. Petite Mutants Induced with HMGR Inhibitors in C. glabrata and C. albicans 

HMGR inhibitors such as statins are reported to produce mitochondrial damage [9], which is sometimes capable of generating toxicity and/or drug-resistant strains. Because the compounds under study target HMGR, mitochondrial damage was assessed as the presence of *petite* mutants (yeasts with respiratory or mitochondrial deficiency). The yeasts were treated with the inhibitors for 72 h, the time point that frank induction of mutants is described to take place. Subsequently, aliquots of yeast growth were inoculated onto a YPD medium and incubated at 37 °C for 72 h. The mutants grew slower and therefore had smaller colonies than the wild strains, as expected due to mitochondrial damage, which results in deficient respiratory and fermentative capacity. Figure 4a,b portray some of the mutants obtained by treating *C. glabrata* (Figure 4a) and *C. albicans* (Figure 4b) with fluconazole, simvastatin, **6b**, **6d** and **6f**.

### 2.6. Evaluation of Mitochondrial Damage and Confirmation of the Petite Phenotype Generated in C. glabrata and C. albicans with HMGR Inhibitors

The *petite* phenotype has a partial or total loss of mtDNA. With partial loss (a lower number of mitochondria), yeast cells can grow well on media with a fermentable carbon source (YPD) and to a lesser extent on media with a non-fermentable carbon source (YPG or YNB + glycerol). With total loss of mtDNA, yeast cells do not grow, even on YPG or YNB + glycerol. To assess the mitochondrial respiratory activity of the wildtype strain and the *petite* mutants with partial and total loss of mtDNA, they were all grown on YPD, YPG and YNB + glycerol (see Figure 5a,b). 

After confirming the phenotype based on the assimilation of the carbon source, the *petite* mutants were further characterized by amplification of mitochondrial genes, using primers previously reported as indicators of a partial or total loss of mtDNA. The pattern of three amplicons (with molecular sizes of approximately 1500, 1250 and 1000 bp) reveals the existence of complete mtDNA, corresponding to the wild strains of *C. glabrata* and *C. albicans* (in lanes 15 and 28, respectively, of Figure 5c,d). The strains with a partial loss displayed a lower number of bands (lanes 3, 6, 9 and 10 for *C. glabrata*; lanes 25 and 27 for *C. albicans*), while those with a total loss did not show any amplicons (lanes 1, 2, 4, 5, 7, 8, 11, 12, 13 and 14 for *C. glabrata*; lanes 16, 18, 19, 21, 22, 23 and 24 for *C. albicans*). The results for each compound are summarized in Appendix A, including evidence of the *petite* mutants (based on yeast growth in media with a fermentable and non-fermentable carbon source) and the presence or absence of mtDNA revealed with PCR. 

The loss of mtDNA (null, partial or total) was detected by staining with 4′,6-diamindino-2-phenylindole (DAPI) and visualization with fluorescence microscopy (Figure 5e,f). The wild strains of *C. glabrata* (CgWT) and *C. albicans* (CaWT) exhibited stained nuclear DNA (yellow arrow) and small dots corresponding to the mtDNA (white arrow). The strains with a partial loss of mtDNA (Cg1 and Ca1) had a lesser number of such dots. In contrast, only nuclear DNA was observed in strains with a total loss of mtDNA (Cg2 and Ca2).

### 2.7. Cloning, Expression, Purification and Detection of CgHMGR in Pichia Pastoris

In order to examine whether the test compounds inhibit the HMGR enzyme of *Candida* ssp., the catalytic portion of CgHMGR was cloned in the pPICZB vector (Figure 6a) for expression in the yeast *P. pastoris*. The gene that codes for the CgHMGR enzyme was found to be integrated into *P. pastoris* using homologous recombination (Figure 6b). The CgHMGR was expressed and purified with nickel affinity chromatography. The molecular mass of the purified enzyme was estimated with an SDS–PAGE analysis to be approximately 48 kDa (Figure 6c), detected using a Western blot with a His probe (Figure 6d).

Since the recombinant HMGR from *C. albicans* is currently being investigated (paper in preparation), only the rec-CgHMGR is shown in this study. The data allow for a comparison of the activity of rec-CgHMGR expressed in a eukaryotic system such as *P. pastoris* with that of the same recombinant enzyme in the *Escherichia coli* system, the latter previously described by our group [22].

### 2.8. Inhibition of Rec-CgHMGR Activity with Simvastatin, α-asarone, ***1a***, ***1b***, ***6g*** and ***7a***

To examine whether the HMGR inhibitors affect the activity of rec-CgHMGR, an inhibition kinetics assay was performed with the two controls (simvastatin and α-asarone) and selected compounds **1a**, **1b** and **7a** to exhibit lower MIC_50_ values and **6g** to inhibit the best growth at 100 µM in the susceptibility profile assays. The data on enzymatic activity were used to determine the percentage of inhibition (Table 3).

### 2.9. Molecular Docking with Selected HMGR Inhibitors

A docking study was carried out with **1a**, **1b**, **6g** and **7a** to explore the affinity and binding mode of the inhibitors for the active site of the CaHMGR and CgHMGR enzymes. Regarding CaHMGR, the interaction energy was better for compounds **1a**, **1b**, **6g** and **7a** (Figure 7) than α-asarone. Compounds **1a**, **1b**, **6g** and **7a** recognized the amino acid residues of the active site of CaHMGR and had lower binding energy values than α-asarone, suggesting that such compounds could be an option in antifungal treatment of *C. albicans*. These data are in accordance with a previous report by our group on CgHMGR [23]. 

## 3. Discussion

Since the increasing resistance of yeasts to antifungal drugs has made certain infections in patients difficult to treat, there is a need to find new drugs with novel molecular targets. Several strains of the genus *Candida* have been used as study models. The availability of the genome of *Candida* has allowed for the identification of new specific targets in pathogenic fungi and for the design of new drugs aimed at these targets [24]. 

Series of test compounds **1a**–**b**, **2a**–**c**, **3a**–**b**, **4a**–**c**, **5a**–**b**, **6a**–**g** and **7a**–**b** were presently designed with the aim of providing a stronger inhibition of this enzyme than that caused by simvastatin. The latter is known to be a competitive inhibitor of HMGR and to inhibit the growth of *C. glabrata* [9,25]. The HMGR enzyme is highly conserved in all fungi and participates in ergosterol synthesis, which is vital for the proper formation of the yeast membrane [26]. 

The current analysis revealed that the HMGRs of *C. glabrata* and *C. albicans* contain a transmembrane domain and a catalytic domain, characteristics of class I HMGRs. Whereas the transmembrane domain has eight segments in *C. glabrata* and nine in *C. albicans*, the catalytic domain is highly conserved. Thus, the test compounds could possibly act as antifungal agents on the HMGR from other *Candida* strains as well by complying with Lipinski’s rules, which are used to evaluate the oral availability of a compound [27].

The in vitro inhibitory effect of the test compounds was evaluated on six strains of *C. glabrata* and *C. albicans*. When determining the MIC of reference compounds (simvastatin and fluconazole) against these strains, a similar phenotype was obtained. Both reference compounds had a MIC_50_ of 8–128 μg/mL for *C. glabrata* strains and greater than 64 μg/mL for *C. albicans* strains. Considering that **1a**, **1b** and **7a** had MIC_50_ values under 256 μg/mL, they may have potential as antifungal agents. Regarding series **1**, the IC_50_ value was lower for **1b** than **1a**. For series **7**, the IC_50_ value was better for **7a** than **7b**.

Derivatives **1a** and **1b** were substituted at the C-4 carbon with polar groups (NO_2_ and OH, respectively), which enhanced their activity. According to the docking simulations between these compounds and the *S. pombe* HMGR enzyme, the C-4 polar groups increase the interaction energy between the hydrogen and the amino acid residues of the active site. In addition, the acetic chain in the compounds of series **1** appears to function in a manner equivalent to the methyl group in series **7** [18]. 

The effect of the test compounds on yeast growth was assessed at 0, 3, 6, 9 and 12 h by performing viable cell counts. Compound **1a** proved to be more lethal than **1b** when each was applied at a concentration of 300 μM. Both compounds were more effective on *C. albicans* than *C. glabrata*. At a concentration of 100 μM, however, the inhibition seemed to decrease as the exposure time increased, a condition possibly related to the development of resistance [28]. The same phenomenon of decreased inhibition with increased exposure time occurred for the lowest concentration of **2a** and **2b** (100 μM). Overall, the inhibition produced with series **2** was similar to that of series **1**. Compound **3a** (but not **3b**) still showed a constant inhibition at 12 h of exposure and a similar inhibition at the concentrations of 100 and 300 μM. The inhibition was greater in *C. albicans* than in *C. glabrata*. Within series **4**, no difference existed in the growth inhibition generated with the compounds, but activity was greater on *C. albicans* than *C. glabrata*. Similarly, there was no difference between the activity of **5a** and **5b**, but they were generally more effective on *C. albicans* than *C. glabrata*. In series **6**, better inhibition was furnished using **6b** compared to **6d**, while the effect of **6g** was notably lower. The inhibitory activity of **6b** was even better than fluconazole and simvastatin, as clearly illustrated in the susceptibility profile in Figure 2. For series **7**, the inhibition was similar for **7a** and **7b**. Hence, compounds in each series demonstrated growth inhibition of the two *Candida* species, as fluconazole and simvastatin are known to do. 

Based on the quantification of membrane sterols [29], all test compounds proved to be capable of affecting sterol biosynthesis in the six strains of *C. albicans* and *C. glabrata*. By acting on the HMGR enzyme, simvastatin was able to diminish ergosterol levels from 23 to 92% in all strains evaluated, as has been found in previous studies [9,30,31,32,33]. Inhibition of ergosterol biosynthesis was also found for the test compounds. Although the inhibitory activity was similar for **1a** and **1b**, it was better for **1a** (>50%) than **1b** (<50%) against *C. glabrata* strains. In the compounds of series **2**, inhibition in most strains was 80–100%. Within series **3**, compounds **3a** and **3b** showed similar activity, evidenced by an inhibition of 18–64% and 12–50% for the three strains of *C. glabrata* and three strains of *C. albicans*, respectively. In series **4**, all three compounds had similar activity, likely due to their acetic chain and polar substituents. Regarding the compounds of series **5**, **5a** exhibited greater inhibition than **5b**, perhaps due to the influence of the position of the chlorine substituent on the interaction with the residues of the catalytic site. Since all compounds in series **6** provided a robust inhibition on ergosterol biosynthesis, the position of the corresponding substituents apparently does not make a difference. Finally, **7a** and **7b** afforded 100% inhibition, independently of the presence or absence of the double bond on the side chain.

The ability of the compounds to induce *petite* mutants was analyzed because the mitochondrial dysfunction generated with such mutants can lead to the development of resistance to treatment in strains normally sensitive to azoles [27]. Furthermore, a relation has been suggested between the emergence of these mutants and toxicity. Taking yeasts as a model, this relation represents indirect evidence of the possible effect of the compounds on human cells [9], considering that the liver damage caused by HMGR inhibitors (e.g., simvastatin) is associated with mitochondrial damage, which is known to trigger apoptosis in liver cells [17].

*Petite* mutants were indeed induced in yeast cells with some of the current test compounds. However, such compounds produced less mutants than fluconazole and simvastatin. The resulting mutant strains were grown on media with glucose (a fermentable carbon source) or glycerol (a non-fermentable carbon source). Yeasts with respiratory deficiency are incapable of growing on media with glycerol and grow at a slower rate on media with glucose than the wild strain [9]. 

By growing the yeasts on distinct media, *petite* mutants could be selected. To confirm the results, the same mutants were subjected to PCR to detect the presence of mitochondrial genes (see Materials and Methods). Subsequently, yeast cells were labeled with DAPI and visualized with fluorescence microscopy to determine the partial or total loss of mtDNA. The results from PCR coincided with the phenotypes observed with yeast growth. 

The relatively low rate of induction of *petite* mutants with the test compounds and the low toxicity of these compounds are in agreement with their structures. Similar modifications in the structure of α-asarone are reported to reduce its mutagenic potential. During the metabolism of α-asarone, an epoxide is produced by an oxidation pathway that is similar to one found in trans-anethole [34]. The epoxide causes the mutagenic and carcinogenic potential of the drug. Consequently, the test compounds developed by mimicry of the α-asarone structure were designed to prevent the formation of an epoxide in order to avoid damage to human cells. To achieve this goal, they had a hydroxyl group on the C-1 carbon of the propanylic chain (eliminating the double bond), as well as saturation of the side chain. 

In order to demonstrate that the compounds act effectively on the HMGR enzyme of *Candida* spp., the cloning and heterologous expression of the HMGR gene of *C. glabrata* were achieved in *P. pastoris*. Accordingly, a 1317 bp amplicon corresponding to the catalytic portion of the HMGR gene was obtained, cloned in the expression vector pPICZB, and transferred to *P. pastoris*. The HMGR gene proved to be responsible for increasing HMGR activity when *P. pastoris* was grown in induction conditions, a finding corroborated after the recombinant enzyme was purified and its activity was inhibited with the test compounds. Hence, the drug-induced inhibition of *C. glabrata* likely stems from the inhibition of the HMGR enzyme. Compounds **1a**, **1b**, **6g** and **7a** turned out to be good inhibitors and therefore a candidate to be developed as a drug for treating opportunistic fungal diseases. The current results are comparable to those described for CgHMGR expressed in *E. coli*, suggesting that the enzymatic activity of CgHMGR and its inhibition with HMGR inhibitors occur independently of the expression system employed [22,23].

The insights provided in the molecular docking studies are also in agreement with those of the in vitro inhibition assays and the heterologous expression of the HMGR gene of *C. glabrata* in *P. pastoris*. In a recent study, it was found that **1a** and **6g** bind to amino acid residues of the active site of CgHMGR with greater affinity than α-asarone, and that **6g** showed the best affinity [23]. The same analysis was conducted presently with CaHMGR, finding greater affinity for **1a**, **1b**, **6g** and **7a** than α-asarone. Thus, the possibility of obtaining the recombinant HMGR protein from *C. albicans*, as was carried out with *C. glabrata*, and using it as a model for the study of these and other new antifungal agents is currently being investigated.

One of the limitations of this study was the ability to decide which compound should be considered the best alternative for inhibiting *C. glabrata* and *C. albicans*. Although compound 6g presented a MIC_50_ value >256, the susceptibility at different concentrations, the ergosterol synthesis and the enzymatic inhibition studies all indicated that it was one of the best compounds. However, this could be due to the well-known phenomenon called “trailing”, in which a strain may be interpreted with the CLSI method as resistant but later identified with other methods (e.g., an animal model) as a sensitive phenotype. On the other hand, it was also necessary to carry out an initial study of toxicity. Accordingly, it was found that the compounds induced *petite* mutants, even though this induction was lower than that found with the current reference compounds. Finally, given the strains and conditions presently tested, it could not be determined whether or not there was synergism of each compound with fluconazole or simvastatin. Hence, this exploratory study requires follow up, including further evaluation of toxicity, synergism and inhibition of the HMGR enzyme. Additionally, the recombinant protein should be obtained in *C. albicans* and other pathogenic fungi in order to show that such compounds are indeed capable of inhibiting the HMGR enzyme in a wide variety of microorganisms.

## 4. Materials and Methods

### 4.1. Strains and Culture Media

*Candida albicans* ATCC10235, CAL18 and CAL30 and *Candida glabrata* CBS138, CGL24 and CGL43 were grown in the YPD medium (1% yeast extract, 2% gelatin peptone and 2% glucose) t (MCD Lab, Mexico) and stored at −70 °C with 50% glycerol (J.T. Baker, Mexico). The MIC was determined on the Sabouraud Glucose Agar medium (SDA: 1% casein enzyme digest, 4% dextrose and 1.5% agar) (MCD Lab, Mexico) and Roswell Park Memorial Institute medium (RPMI-1640) with the MOPS (Sigma-Aldrich, St. Louis, MO, USA) regulator at a final concentration of 0.125 mg/mL, adjusted to pH 7 ± 0.1. The wild and *petite* mutant strains were maintained with the YPD medium (1% yeast extract, 2% peptone from casein and 2% dextrose) (J.T. Baker, Mexico). They were grown with YPG (1% yeast extract, 2% peptone from casein and 2% glycerol), YNB + D (0.67% yeast nitrogen base and 2% dextrose) and YNB + G (0.67% yeast nitrogen base and 2% glycerol) (J.T. Baker, Mexico). For the solid medium, 2% bacteriological agar was added. The media were sterilized by autoclaving at 121 °C for 15 min, except for the minimum YNB medium. The vector pPICZB (Easy Select *Pichia* Expression Kit, Invitrogen, Waltham, MA, USA) was used for the cloning and expression of the HMGR gene from *C. glabrata*. The pPICZB-CgHMGR construct was expressed in the *P. pastoris* strain X-33 (Invitrogen, Waltham, MA, USA), and the *E. coli* strain DH10b was used for the plasmid construct and propagation. 

### 4.2. Compounds

The structures of simvastatin, fluconazole (the reference compounds serving as positive controls of inhibition), α-asarone and the chemical structures of test compounds are illustrated in Figure 2. Simvastatin and fluconazole were purchased from (Sigma-Aldrich, St. Louis, MO, USA) while the test compounds were synthesized as previously described [18,19]. 

### 4.3. Determination of Physicochemical Properties of Simvastatin, Fluconazole, α-Asarone and the Series of Test Compounds ***1a***–***b
***, ***2a***–***c***, ***3a***–***b***, ***4a***
–***c***, ***5a***–***b***, ***6a***–***g*** and ***7a***–***b***

The physicochemical and toxicological properties of 2-amino-3-cyano-4*H*-chromenes 4a–o and 6a–h were determined on the OSIRIS DataWarrior V4.7.2 program (https://openmolecules.org/datawarrior/) accessed on 17 November 2023 [27]. 

### 4.4. Bioinformatic Analysis and Topology of the HMGR of C. glabrata and C. albicans

The sequences of the gene encoding HMGR of *C. glabrata* and *C. albicans* were searched and retrieved in the NCBI database (http://www.ncbi.nim.nih.gov) accessed on 2 January 2023. The access number for CgHMGR is XP_449268 and for CaHMGR, it is XP_713681. The search for the ORF and molecular weight of each of these sequences was performed on the DNAMAN v. 3.0 program [35]. The prediction of transmembrane regions (topology) and the isoelectric point (Ip) was made on the SOSUI server (http://bp.nuap.nagoya-u.ac.jp/sosui/) accessed on 2 January 2023 [36]. The probable subcellular location was predicted with PSORTII (http://psort.hgc.jp/form2.html) accessed on 2 January 2023 [37]. “Motif” sequences (consensus sequences representing activity, active site, post-translational modification site, etc.) were predicted with the PrositeScan database, available on the server http://www.expasy.org accessed on 2 January 2023 [38].

### 4.5. Determination of the MIC of HMGR Inhibitors for C. glabrata and C. albicans

The MIC was established by means of the CLSI M27-A3 microdilution method [39]. After the yeast strains were grown for 24 h on SDA plates (MCD Lab, Mexico), 1 mm colonies of the inoculum were taken and suspended in a tube with a saline solution (NaCl, 0.85%), adjusting the optical density (OD) to 0.5 McFarland (equivalent to 0.125 Abs at 525 nm) with a nephelometer. The inoculum was diluted (1:20) in the RPMI-1640 medium and then diluted once again (1:50) in the same medium. Each compound was dissolved in DMSO (J.T. Baker, Mexico) at 100 different concentrations (100× stock solutions). Dilutions were then made in the RPMI-1640 medium to obtain 10× solutions, followed by another dilution to produce 2× solutions, according to the table in the CLSI guidelines. Subsequently, the rows of 96-well plates were filled from wells 1–10 with 100 μL of each compound, from the highest to lowest concentration. Wells 11 and 12 were filled with 100 and 200 μL of 2% RPMI-DMSO (Sigma-Aldrich, St. Louis, MO, USA) (the growth and sterility control, respectively). Columns 1–11 were inoculated with 100 μL of the yeast suspension, followed by incubation of the plates at 37 °C for 48 h. Upon completion of incubation, each microplate was shaken for 30 s to observe whether the growth had decreased, and the absorbance was read at 540 nm on an ELISA microplate reader. The lowest concentration of the test compounds capable of completely inhibiting yeast growth was considered as the MIC. The result was compared between each compound and the untreated control. The MIC_50_ is the lowest concentration of an antifungal agent capable of reducing the optical density value of the growth control by 50%. The CLSI has established categories for azoles: sensitive (<8 µg/mL), SDD (16–32 µg/mL) and resistant (>64 µg/mL).

### 4.6. Effect of HMGR Inhibitors on the Viability and Growth of C. glabrata and C. albicans 

Yeasts (2–5 × 10^3^ cells/mL) were grown in a minimal medium in the presence of 0, 100, 300 or 600 μM of each compound. Aliquots were collected at 0, 3, 6, 9 and 12 h, and the corresponding dilutions were made before spreading 50 μL over YPD agar. After incubation at 37 °C for 24 h, the colonies were counted. The growth inhibition value (%) was calculated with the following formula: [1 − (suspension of a colony with a particular agent/suspension of a colony without any agent)] × 100 [40]. At 12 h of treatment, the cell suspension was dripped onto YPD agar and the growth was observed [41].

### 4.7. Effect of HMGR Inhibitors on Ergosterol Biosynthesis in C. glabrata and C. albicans

The assays to evaluate the inhibition of ergosterol synthesis were carried out in accordance with a previous report [29]. From strains grown for 48 h in the YPD medium (MCD Lab, Mexico), a colony was taken to adjust the inoculum to 0.125 As595 and this suspension was diluted (1:100). After placing 100 μL of the resulting suspension in flasks containing 5 mL of the YNB medium, the solution was inoculated with 100 μM of the compounds and incubated at 37 °C for 16 h under constant shaking. Stationary phase cells were centrifuged at 6000 g for 5 min and then washed twice with sterile distilled water. Subsequent to determination of its wet weight, the pellet was suspended in 3 mL of a 25% alcoholic potassium hydroxide solution, and the solution was mixed in a vortex for 1 min. Cell suspensions were transferred to capped borosilicate tubes and incubated at 85 °C for 1 h, and then allowed to cool to room temperature for 1 h. Sterol extraction was performed by adding 1 mL of sterile distilled water and 3 mL of n-heptane followed by vigorous mixing in a vortex for 3 min. The heptane (J.T. Baker, Mexico) layer was transferred to a borosilicate tube with a screw cap and stored at −20 °C for 24 h. To examine the level of sterols, a 1:5 dilution was made in 100% ethanol, and the absorption spectrum was measured in the 240–300 nm range on a Varian Cary spectrophotometer (50 conc). In the absorption spectrum, ergosterol and late sterol intermediates (24DHE) display a characteristic curve of four peaks. The absence of detectable ergosterol in the extracts is indicated by a flat line. A decrease in the height of the absorbance peaks corresponds to a lower concentration of ergosterol. The content of ergosterol was calculated as a percentage relative to the wet weight of the cells with the following equation:
$$\begin{array}{l} \%\;\mathrm{Ergosterol}+\%\;\text{24DHE}=[(\mathrm{Abs}\;281.5/290)\times F]/\mathrm{pellet}\;\mathrm{weight}\\ \%\;\text{24DHE}=[(\mathrm{Abs}\;230/5189)\times F]/\mathrm{pellet}\;\mathrm{weight}\\ \%\;\mathrm{Ergosterol}=[\%\;\mathrm{ergosterol}+\%\;\text{24DHE}]-\%\;\text{24DHE} \end{array}$$
where F is the dilution factor in ethanol, 290 is the value of E in percent per cm^3^ of crystalline ergosterol and 518 is the value of E in percent per cm^3^ of 24DHE.

### 4.8. Petite Mutants of C. glabrata and C. albicans Induced with the Test Compounds 

*Petite* mutants were induced by exposing *C. glabrata and C. albicans* to 100 μM of the test compounds for 72 h. Upon completion of the incubation, the necessary serial dilutions were made to assess the effect of each compound. On plates containing YPD agar (MCD Lab, Mexico), one of the last two dilutions was added (using 50 µL of the inoculum), followed by incubation at 37 °C for 24–48 h, and finally the performance of viable counts in duplicate. Wild yeast colonies and probable *petite* mutants (small colonies) were morphologically identified by comparing them to colonies grown on the growth control plate (without inhibitors).

### 4.9. Phenotypic Characterization of Selected Petite Mutants

For the verification of the phenotype of *petite* mutants, the CBS138 strain of *C. glabrata* and the ATCC10231 strain of *C. albicans* were grown in the presence of fluconazole, simvastatin (Sigma-Aldrich, St. Louis, MO, USA) and the test compounds at 37 °C for 72 h. The small colonies designated as *petite* mutants were seeded in media with different carbon sources. Two media contained fermentable substrates (YPD and YNB + dextrose) (MCD, Lab, Mexico) and two contained non-fermentable substrates (YPG and YNB + glycerol) (MCD, LaB, Mexico). The colonies that are *petite* respiratory mutants have mtDNA damage and thus are only able to grow on media with a fermentable carbon source since they cannot generate energy by means of the respiratory chain. 

### 4.10. Genotypic Characterization of Selected Petite Mutants

The pattern of bands for the genotypic characterization of the *petite* mutants was provided with amplification of the ribosomal DNA of fungal mitochondrial genes using the PCR technique with ML1 (5′-GTACTTTTGCATAATGGGTCAGC-3′) and ML6 primers (5′-CAGTAGAAGCTGCATAGGGTC-3′), as described in the literature [28]. Genomic DNA of the *petite* mutants served as a template.

### 4.11. Evaluation of Loss of mtDNA in Petite Mutants

The *petite* mutants obtained were grown in YPD broth and washed with a PBS solution with 0.1% triton (J.T. Baker, Mexico). After adding DAPI (Sigma-Aldrich, St. Louis, MO, USA) to a final concentration of 1 μg/mL, the mutants were incubated on ice for 10 min. The labeling was examined at an excitation wavelength of 359 nm and an emission wavelength of 461 nm with fluorescence microscopy on a Nikon Eclipse E800 microscope (Nikon, Minato City, Tokyo, Japan), utilizing a 10 × 25 eyepiece and 100× objective.

### 4.12. Heterologous Expression of the CgHMGR Gene in P. pastoris

The CgHMGR nucleotide sequence (XM_449268.1) was identified through a BLASTp analysis. The coding region for the catalytic region of CgHMGR was amplified using PCR with the following primers: Fwd*Pml*1 5′-TCC CACGTG ATGGTCTCGTTGGTTGGTTATCCACGGTAAGC-3′ and Rev*Sac*II 5′-AGG CCGCGG AGACTTGATGCAAATTTTTGAACCTTCTTCCAATC-3′. The *Pml*1 and *Sac*II restriction sites, respectively, were included (underlined in the sequence). The amplicon was digested with *Pml*I and *Sac*II restriction enzymes and purified with the Zymoclean Gel DNA Recovery Kit (ZymoResearch, Irvine, CA, USA), following the manufacturer’s instructions. The purified product was digested and cloned into the *Pml*1 and *Sac*II sites of the pPICZB multiple cloning site to generate plasmid pPICZB-CgHMGR, based on previously described techniques [42]. *P. pastoris* was transformed and linearized with *Pme*I. Transformant clones were selected on the YPD medium supplemented with zeocin (Thermo Scientific, Waltham, MA, USA) (YPD, 100 µg/mL of zeocin and 1 M sorbitol), according to the manufacturer’s instructions. The *P. pastoris* transformants were confirmed with PCR amplification of the CgHMGR gene, employing the same primers involved in cloning as well as the AOX forward and reverse primers supplied with the kit. *P. pas*toris X-33 genomic DNA served as the negative control.

### 4.13. Expression, Purification and Detection of the Recombinant CgHMGR

P. pastoris wildtype and transformant strains were grown in 50 mL of the glycerol-based yeast medium (1.34% YNB without amino acids, 4 × 10^−5^% biotin, 10 mM of potassium phosphate buffer, and 2% glycerol, pH 6.0) (Thermo Scientific, Waltham, MA, USA) at 30 °C in an agitated water bath until reaching OD_600_ (~2.0). The cells were harvested and resuspended in 100 mL of the same medium supplemented with 0.5% methanol (*v*/*v*) instead of glycerol and then incubated for 2 days. Methanol was added every day at the same concentration. The cell-free extract (CFE) was obtained by mechanical stirring with glass beads, and subsequently the cells were resuspended in a phosphate regulator. The CFE was purified to afford CgHMGR using nickel affinity chromatography with NTA-agarose (Thermo Scientific, Waltham, MA, USA) following the manufacturer’s protocol. Fractions were collected for protein determination by using the Lowry assay [43], then analyzed in 10% SDS–PAGE and stained with Coomassie R-250 [44]. Selected fractions were frozen at −70 °C until needed. The recombinant CgHMGR protein was detected using a Western blot with the His-Probe-HRP (Invitrogen, Waltham, MA, USA). 

### 4.14. Evaluation of Enzymatic Activity and Enzyme Inhibition 

The HMGR enzymatic activity and inhibition assays were performed in triplicate, as described previously [45]. They are based on a decrease in absorbance of NADPH measured at 340 nm. NADPH (Sigma-Aldrich, St. Louis, MO, USA) is converted into its oxidized form by the catalytic subunit of HMGR in the presence of HMG-CoA. For the evaluation of enzymatic activity, a reaction mixture containing 130 µM of HMG-CoA, 1–64 µM of (R/S)-HMG-CoA, 50 mM of Tris-HCl (pH 8.0), 130 µM of NADPH and 10 µg/mL (Sigma-Aldrich, St. Louis, MO, USA) of the enzyme was incubated at 37 °C for 1 h. NADPH oxidation was monitored every 10 s over a period of 10 min in a BioPhotometer^®^ 8452 spectrophotometer at 340 nm (Eppendorf). The inhibition assay was conducted by incubating the enzyme with 100 μM of simvastatin, α-asarone, 1a, 1b, 6g, 7a and employing DMSO as the control. CgHMGR activity was expressed in units, where one unit is the quantity capable of converting 1.0 μm of NADPH to NADP^+^ per min at 37 °C.

### 4.15. Molecular Docking Study

A previously validated docking study was conducted in order to provide insights into the binding mode of 1a, 1b, 6g, 7a, α-asarone and simvastatin on the CaHMGR enzyme. The docking simulations were carried out on Autodock version 4.0 [46]. The model of CaHMGR was obtained with the Modeller program, [47] utilizing the crystal sequence of human HMGR (hHMGR) as a template. The structure of hHMGR was deposited in the Protein Data Bank (PDB: 1DQ8). The structures of the ligands were drawn with MarvinSketch (https://www.chemaxon.com/products/marvin/marvinsketch/) accessed on 6 March 2023 and converted into the MOL2 file format on the Open Babel GUI program [48]. For the preparation of docking, the parameters were estimated in AutoDock Tools. Polar hydrogens were added to the protein. The grid dimensions were set at 98 × 68 × 62 Å^3^ with points separated by 0.375 Å, and the following grid center: X = −13.305, Y = 19.729 and Z = 22.776. Random starting positions, orientations and torsion were established for all ligands. Default values of translation, quaternation and torsion steps were adopted for the simulation. The hybrid Lamarckian Genetic Algorithm was applied for minimization, using default parameters. The number of docking runs was 100. Docking results were analyzed in AutodockTools and edited in Discovery 4.0 Client (San Diego, CA, USA) [49]. 

## 5. Conclusions

In conclusion, according to the results, series of test compounds **1a**–**b**, **2a**–**c**, **3a**–**b**, **4a**–**c**, **5a**–**b**, **6a**–**g** and **7a**–**b** (proposed as inhibitors of the HMGR enzyme of *Candida*) may act as better antifungal agents than commercial drugs. The test compounds could outperform such drugs in the growth reduction in *C. glabrata* and *C. albicans* strains, the decrease in ergosterol synthesis and viability and the more limited generation of petite mutants. Moreover, the model of the recombinant version generated presently will be instrumental for testing new compounds in the future, and thus be helpful in the search for more effective antifungal molecules to treat infections with different species of *Candida* yeasts.

## Figures and Tables

**Figure 1 ijms-24-16868-f001:**
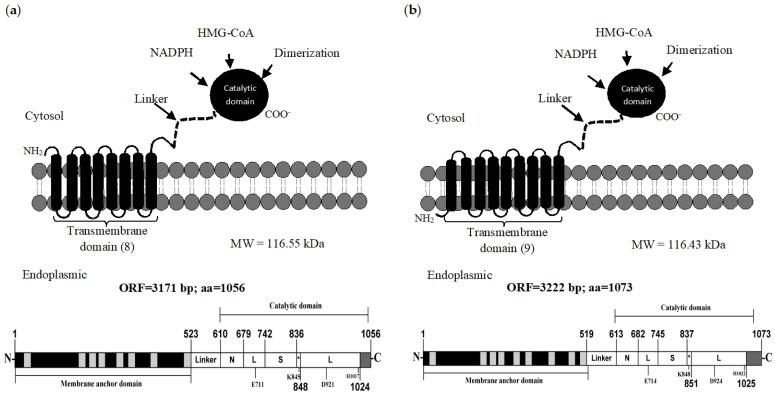
Model and organization of the HMGRs of *C. glabrata* and *C. albicans*. The secondary structures of the HMGRs of *C. glabrata* (CgHMGR) (**a**) and *C. albicans* (CaHMGR) (**b**) reveal a topology similar to that of class I HMGRs. Accordingly, there is a transmembrane domain at the N-terminal (with a distinct number of segments for each enzyme) and a catalytic domain in the C-terminal consisting of several subdomains, including an NADPH binding domain, an HMG-CoA binding domain and a dimerization domain. The asterisk shows the cis-loop connects the HMG-CoA-binding region with the NADP-H-binding region.

**Figure 2 ijms-24-16868-f002:**
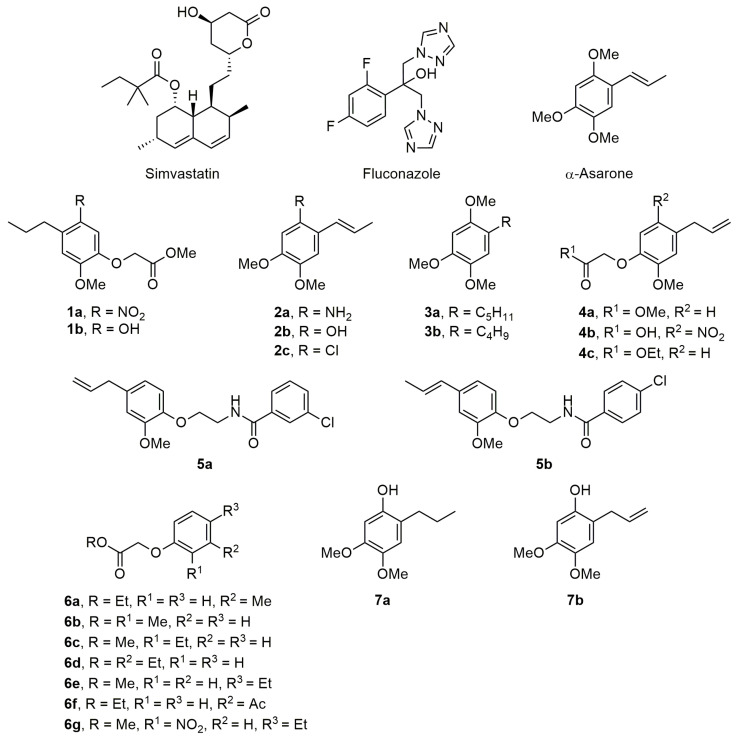
Structures of simvastatin, fluconazole, α-asarone and the series of test compounds **1a**–**b**, **2a**–**c**, **3a**–**b**, **4a**–**c**, **5a**–**b**, **6a**–**g** and **7a**–**b**.

**Figure 3 ijms-24-16868-f003:**
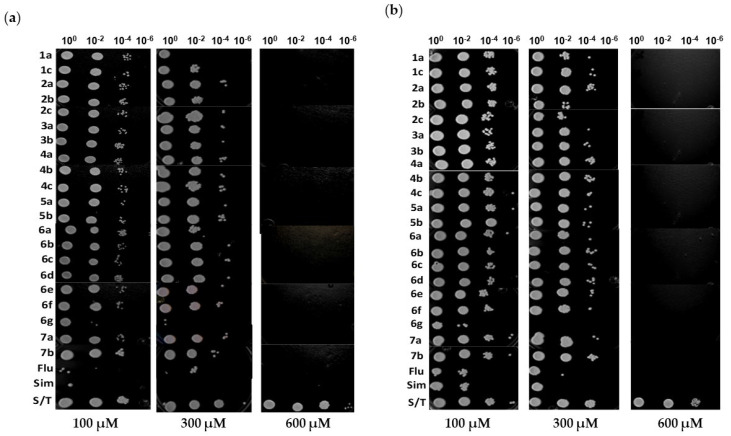
Susceptibility profile of *C. glabrata* (**a**) and *C. albicans* (**b**) to HMGR inhibitors. The yeasts were treated with 100, 300 and 600 μM of the compounds of the series **1**, **2**, **3**, **4**, **5**, **6** and **7** for 24 h. Serial dilutions were made with the resulting growth and 5 μL was dripped onto YPD plates, which were photographed after 48 h of incubation at 37 °C.

**Figure 4 ijms-24-16868-f004:**
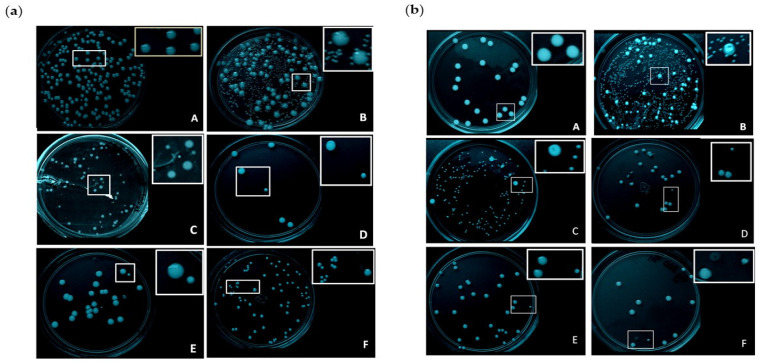
Induction of *petite* mutants in *C. glabrata* (**a**) and *C. albicans* (**b**). The panels depict yeast cells without treatment (**A**) and those treated with fluconazole (**B**), simvastatin (**C**), **6d** (**D**), **6f** (**E**) and **6b** (**F**). The cells were grown in the presence of the compounds for 72 h, followed by inoculation of aliquots of yeast growth on YPD agar, to be incubated at 37 °C for 72 h. Whereas the colonies were large for the wild strain (without mitochondrial damage), they were small for the *petite* mutants (with mitochondrial damage). The two phenotypes are shown in detail in the boxes portraying close-ups of growth on agar.

**Figure 5 ijms-24-16868-f005:**
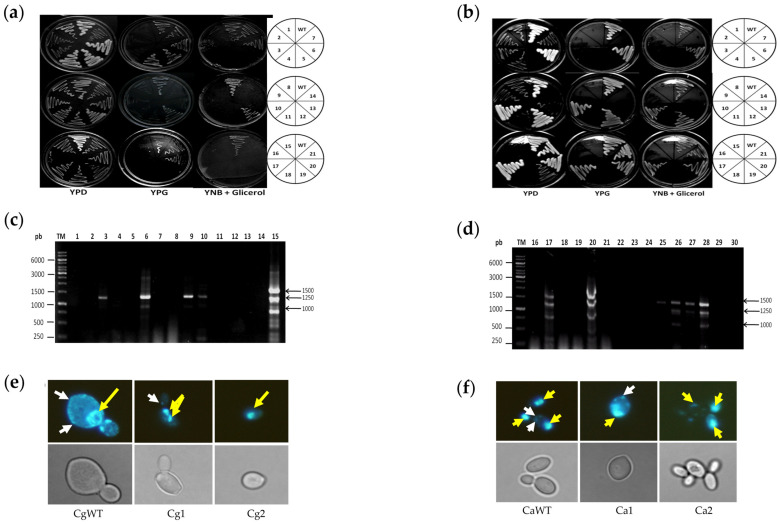
Evaluation of the loss of mitochondrial function in *C. glabrata* and *C. albican*s and the presence of the *petite* genotype. (**a**,**b**) The strains of *C. glabrata* CBS138 (**a**) and *C. albicans* ATCC10231 (**b**) were grown at 37 °C for 72 h in media containing the compounds, and the small colonies were selected as *petite* mutants. The mutants were inoculated onto plates with YPD, YPG or YNB + glycerol and incubated for 72 h. The growth on the different media and the diagram of each plate are illustrated, indicating the designated number for each mutant obtained. Whereas growth of the wild strain (WT) can be observed on all three media, strains with a total loss of mtDNA did not show growth on any of these media. There was less growth on media with glycerol for the mutants with a partial loss of mtDNA. (**c**,**d**) Electrophoretic pattern of mtDNA amplicons in *C. glabrata* (**c**) and *C. albicans* (**d**): the wild strain of *C. glabrata* (lane 15) and *C. albicans* (lane 28), and the *petite* mutants of *C. glabrata* (lanes 1–14) and *C. albicans* (lanes 16–27). A molecular size marker of 1 Kb (TM) and controls without DNA (lanes 29 and 30). (**e**,**f**) Loss of mtDNA in *C. glabrata* and *C. albicans* due to treatment with HMGR inhibitors. After treatment, cells were stained with DAPI (2 µg/mL) and visualized with fluorescence microscopy. *C. glabrata* and *C. albicans* without treatment (CgWT and CaWT, respectively), with a partial loss of mtDNA (Cg1 and Ca1), and with a total loss of mtDNA (Cg2 and Ca2). The figure portrays nuclear DNA marked with yellow arrows and mtDNA with white arrows. 100× objective.

**Figure 6 ijms-24-16868-f006:**
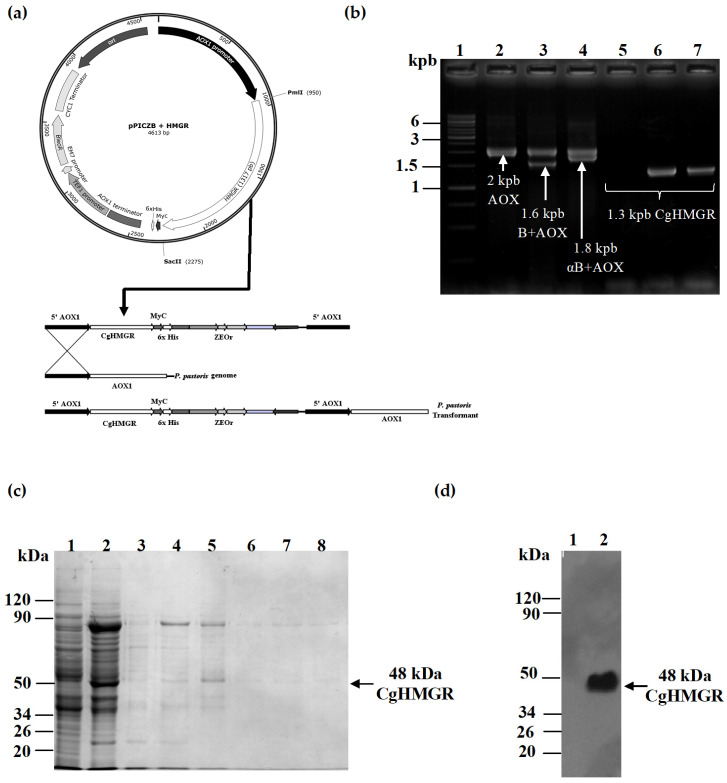
Cloning, expression, purification and detection of CgHMGR. (**a**) Representation of the homologous recombination in *P. pastoris*. (**b**) Gel electrophoresis confirmed the presence of the homologous recombinant HMGR gene from *C. glabrata* in *P. pastoris*: lane 1, molecular weight marker; lane 2, amplification of the AOX gene in *P. pastoris*; lanes 3 and 4, testing for the homologous recombination of pPIZCB and pPICZαB in *P. pastoris* with AOX oligos; lane 5, negative amplification of the CgHMGR gene in *P. pastoris* X-33; lanes 6 and 7, testing for the homologous recombination of pPIZCB and pPICZαB in *P. pastoris* with CgHMGR oligos. (**c**) Electrophoretic analysis of the expression and purification of the soluble fraction of CgHMGR with nickel affinity chromatography (a molecular weight marker is shown on the left of each gel): lane 1, cellular extraction from uninduced cells; lane 2, cellular extraction from methanol-induced cells; lanes 3–8, eluted fractions. The black arrow points to the band corresponding to CgHMGR, with the expected molecular size. (**d**) Identity of the purified protein was verified using Western blot with the histidine probe: lane 1, marker; lane 2, purified CgHMGR.

**Figure 7 ijms-24-16868-f007:**
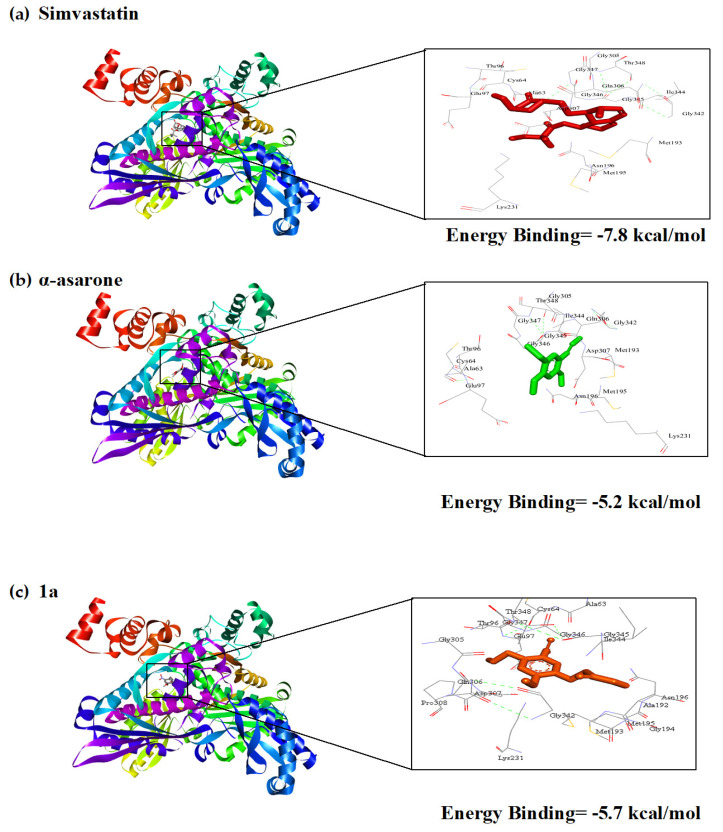
Examination of the binding mode and interaction energy for simvastatin (**a**), α-asarone (**b**), **1a** (**c**), **1b** (**d**), **6g** (**e**) and **7a** (**f**) with the catalytic portion of CaHMGR. The dimeric protein CaHMGR is portrayed with a flat ribbon. The interactions of each compound with the catalytic portion of CaHMGR are illustrated in a 3D model, which shows the amino acids from the active site of the enzyme (Glu97, Lys321 and Asp307) that are involved in the interaction.

**Table 1 ijms-24-16868-t001:** The 50% minimum inhibitory concentration (MIC_50_) of HMGR inhibitors for three *C. glabrata* and three *C. albicans* strains.

	Strain	MIC_50_ *C. glabrata* Strains			MIC_50_ *C. albicans* Strains		
Compound		CBS138	CGL24	CGL43	ATCC 10231	CAL18	CAL30
**Fluconazole**		8	32	128	4	16	128
**Simvastatin**		8	32	128	128	256	>256
**1a**		**32**	256	256	256	**128**	256
**1b**		>256	>256	>256	**0.5**	**0.5**	256
**2a**		>256	>256	>256	>256	>256	>256
**2b**		>256	>256	>256	>256	>256	>256
**2c**		>256	>256	>256	>256	>256	>256
**3a**		>256	>256	>256	>256	>256	>256
**3b**		>256	>256	>256	>256	>256	>256
**4a**		>256	>256	>256	>256	>256	>256
**4b**		>256	>256	>256	>256	>256	>256
**4c**		>256	>256	>256	>256	>256	>256
**5a**		>256	>256	>256	>256	>256	>256
**5b**		>256	>256	>256	>256	>256	>256
**6a**		>256	>256	>256	>256	>256	>256
**6b**		>256	>256	>256	>256	>256	>256
**6c**		>256	>256	>256	>256	>256	>256
**6d**		>256	>256	>256	>256	>256	>256
**6e**		>256	>256	>256	>256	>256	>256
**6f**		>256	>256	>256	>256	>256	>256
**6g**		>256	>256	>256	>256	>256	>256
**7a**		**64**	**64**	**64**	**64**	**64**	**32**
**7b**		256	256	256	>256	256	>256

**Table 2 ijms-24-16868-t002:** Decrease in ergosterol biosynthesis in *C. glabrata* and *C. albicans* generated by HMGR inhibitors.

Compound (100 μM)	Average Ergosterol Content of Cells Grown in Media with HMGR Inhibitors (100 µM)					
	*Candida glabrata*			*Candida albicans*		
	CBS138	CGL 24	CGL 43	ATCC10231	CAL 18	CAL 30
**Control**	0.074 ± 0.001	0.085 ± 0.0009	0.032 ± 0.001	0.036 ± 0.001	0.030 ± 0.0004	0.037 ± 0.0002
**Simv**	0.007 ± 0.003(65)	0.025 ± 0.002(92)	0.023 ± 0.002(31)	0.027 ± 0.003(23)	0.018 ± 0.003(40)	0.018 ± 0.002(73)
**1a**	0.006 ± 0.003(53)	0.014 ± 0.003(0)	0.01 ± 0.002(35)	0.015 ± 0.002(0)	0.002 ± 0.001(98) ***	0.001 ± 0(97) *
**1b**	0.012 ± 0.002(32)	0.013 ± 0.001(17)	0.016 ± 0.007(38)*	0.018 ± 0.003(0)	0.002 ± 0(97) ***	0.003 ± 0.001(87) *
**2a**	0 ± 0.001(100) *	0 ± 0.01(100)	0 ± 0.009(100) ***	0.019 ± 0.003(0)	0 ± 0.005(100) ***	0.001 ± 0.001(90) *
**2b**	0 ± 0.008(100) *	0 ± 0.009(100)	0 ± 0.0004(100) ***	0 ± 0.001(100)	0.032 ± 0.03(0) ***	0.006 ± 0.001(45)
**2c**	0 ± 0(100) *	0 ± 0.002(100)	0 ± 0.004(100) ***	0.002 ± 0.0003(82)	0 ± 0.001(100) ***	0.001 ± 0.0002(84)
**3a**	0.012 ± 0.003(24)	0.012 ± 0.002(4)	0.005 ± 0.001(64)	0.013 ± 0.001(0)	0.026 ± 0.007(18)	0.031 ± 0.006(28)
**3b**	0.007 ± 0.002(49)	0.012 ± 0.002(0)	0.007 ± 0.0009(54)	0.02 ± 0.008(0)	0.1 ± 0.091(12)	0.025 ± 0.011(28)
**4a**	0.003 ± 0.0003(83) *	0.004 ± 0.002(70)	0.002 ± 0.0003(82) ***	0 ± 0.001(100)	0.004 ± 0.0004(70) ***	0 ± 0.0003(100) ***
**4b**	0.003 ± 0.001(74) *	0.004 ± 0.002(54)	0.003 ± 0.0003(85) ***	0 ± 0.001(100)	0.003 ± 0.002(86) ***	0.002 ± 0.0008(97) ***
**4c**	0.006 ± 0.002(57)	0.007 ± 0.002(35)	0.003 ± 0.001(77) ***	0.004 ± 0.002(59)	0.003 ± 0.002(79) ***	0.002 ± 0.0008(79)
**5a**	0 ± 0(100) *	0 ± 0.0001(100)	0.009 ± 0.004(42) ***	0 ± 0.0008(100)	0 ± 0.002(100) ***	0 ± 0.003(100) ***
**5b**	0.005 ± 0.001(60)	0.005 ± 0.002(54)	0.003 ± 0.0005(76) ***	0.0009 ± 0.003(92)	0.009 ± 0.005(32) ***	0.003 ± 0.002(71)
**6a**	0 ± 0.0006(100) *	0 ± 0.0005(100)	0 ± 0.0003(100) ***	0 ± 0.001(100)	0 ± 0.0003(100) ***	0.0008 ± 0.001(92) ***
**6b**	0 ± 0.004(100) *	0 ± 0.001(100)	0 ± 0.001(100) ***	0 ± 0.004(100)	0 ± 0.0005(100) ***	0.002 ± 0.0008(79)
**6c**	0 ± 0.005(100) *	0 ± 0.005(100)	0 ± 0.0059(100) ***	0 ± 0.0002(100)	0 ± 0.27(100) ***	0 ± 0.008(97) ***
**6d**	0 ± 0.011(100) *	0 ± 0.03(100)	0 ± 0.0007(100) ***	0 ± 0.004(100)	0 ± 0.002(100) ***	0 ± 0.0007(100) ***
**6e**	0.0001 ± 0.0002(99) *	0.025 ± 0.01(57)	0.06 ± 0.006(67) ***	0.055 ± 0.009(96) *	0.007 ± 0.0013(92) ***	0 ± 0.0012(100) ***
**6f**	0.0048 ± 0.003(86) *	0.016 ± 0.003(45)	0.017 ± 0.003(55)	0.002 ± 0.0006(97)	0.006 ± 0.0002(92) ***	0.013 ± 0.002(58)
**6g**	0.007 ± 0.002(74) *	0.009 ± 0.003(68)	0.011 ± 0.006(70)	0.003 ± 0.01(95)	0.008 ± 0.001(90) ***	0.031 ± 0.009(90) ***
**7a**	0 ± 0.0005(100) *	0 ± 0.001(100)	0 ± 0.0005(100) ***	0 ± 0009(100)	0.0004 ± 0.002(97) ***	0 ± 0.0005(100) ***
**7b**	0 ± 0.001(100) *	0 ± 0.002(100)	0 ± 0.001(100) ***	0 ± 0.0008(100)	0 ± 0.004(100) ***	0 ± 0.001(100) ***

In parentheses, the following is shown: the percentage of the reduction compared to the content of cells grown without treatment ± standard deviation. An asterisk or asterisks denote a significant difference between the inhibition of yeast cells with a given test compound and with simvastatin (Simv) (* *p* < 0.05, *** *p* < 0.0001).

**Table 3 ijms-24-16868-t003:** Effect of simvastatin, α-asarone, **1a**, **1b**, **6g** and **7a** on the activity of CgHMGR.

Concentration (100 µM)	Inhibition (% of Relative Activity) *
**HMGR w/o inhibitor**	0
**simvastatin**	97 ± 2.51 *
**α-asarone**	65 ± 2.64 *
**1a**	73 ± 1.52 *
**1b**	89 ± 1.24 *
**6g**	86 ± 3.21 *
**7a**	66 ± 2.08 *

* Relative activity, expressed as a percentage of the activity of CgHMGR in the absence of any inhibitor (considered as 100%). Values were the result of the average of three independent replicates ± standard deviation (SD), * *p* < 0.001, Student’s *t* test (compared with 0 μM inhibitor).

## Data Availability

The data supporting this article will be shared upon reasonable request to the corresponding author.

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
