# Peer review of "Inhibitors of 3-Hydroxy-3-methylglutaryl Coenzyme A Reductase Decrease the Growth, Ergosterol Synthesis and Generation of petite Mutants in Candida glabrata and Candida albicans"

_ijms, 2023, doi:10.3390/ijms242316868_

Round 1

Reviewer 1 Report

Comments and Suggestions for Authors

The manuscript entitled "Inhibitors of 3-hydroxy-3-methyl glutaryl coenzyme A reductase (HMGR) decrease growth, ergosterol synthesis, and petite mutants in Candida glabrata and Candida albicans, also reduce the enzymatic activity of CgHMGR expressed in Pichia pastoris" authored by Dulce Andrade-Pavón et al. is a very well structured and written study 

The aim and the novelty are well covered by the authors, the results are obtained based on suitable and standardized protocols

Two recommendations for the authors:

- reduce the length of the title

- add limitations of the study 

Author Response

Reviewer 1:

Comments and Suggestions for Authors

The manuscript entitled "Inhibitors of 3-hydroxy-3-methyl glutaryl coenzyme A reductase (HMGR) decrease growth, ergosterol synthesis, and petite mutants in Candida glabrata and Candida albicans, also reduce the enzymatic activity of CgHMGR expressed in Pichia pastoris" authored by Dulce Andrade-Pavón et al. is a very well structured and written study 

The aim and the novelty are well covered by the authors, the results are obtained based on suitable and standardized protocols

Two recommendations for the authors:

- reduce the length of the title

- add limitations of the study 

Response 1 at reviewer 1:  We appreciate the reviewer's comment. We reduce the title to: “Inhibitors of HMGR decrease the growth, ergosterol synthesis and generation of petite mutants in Candida glabrata and Candida albicans”.

Response 2 at reviewer 1:  We add the limitations of the study in the discussion section.

Now lines 403-418

“One of the limitations of this study was the ability to decide which compound should be considered the best alternative for inhibiting C. glabrata and C. albicans. Although compound 6g presented a MIC50 value >256, the susceptibility at different concentrations, the ergosterol synthesis, and the enzymatic inhibition studies all indicated that it was one of the best compounds. However, this could be due to the well-known phenomenon called "trailing", in which a strain may be interpreted by the CLSI method as resistant but later identified by other methods (e.g., an animal model) as a sensitive phenotype. On the other hand, it was also necessary to carry out an initial study of toxicity. Accordingly, it was found that the compounds induced petite mutants, even though this induction was lower than that found with the current reference compounds. Finally, given the strains and conditions presently tested, it could not be determined whether or not there was synergism of each compound with fluconazole or simvastatin. Hence, this exploratory study requires follow-up, including further evaluation of toxicity, synergism, and inhibition of the HMGR enzyme. Additionally, the recombinant protein should be obtained in C. albicans and other pathogenic fungi in order to show that such compounds are indeed capable of inhibiting the HMGR enzyme in a wide variety of microorganisms”

Reviewer 2 Report

Comments and Suggestions for Authors

The work submitted to me for review, concerning the search for new antifungal substances for specific use in Candida infections, concerns extremely important and current topics. A very good direction of research in this field is to pay attention to the inhibition of the ergosterol synthesis and consequently reduction of ergosterol levels by HMGR inhibitors, and the use of simvastatin as a reference substance in provided determinations. Although the results obtained are not impressive in all the presented areas, they are nevertheless important for the further effective development of this type of drugs.

Conceptually, the work is good planned, and it was made and presented very well. The scope of the research conducted is appropriate to draw the presented conclusions. Even though the tested compounds are not new substances, or perhaps because of it, this work is more biological and biochemical in nature and should be assessed in this form and scope. However, from a chemical point of view, I have some doubts about the use of the term "alpha-asarone analogues". The tested compounds are quite distant and highly differentiated analogues of azarone, which is considered as a standard substance. It is probably better to use everywhere the phrase used in the caption to Fig.2. Among typical editorial errors, I would like to point out the following:

- line 108 should be "SDD" instead of "SSD".

- in Table 1, in its header I would suggest swapping the words "Compound" and "Strain" and at the same time reversing the position of the diagonal line between these words.

- in the same Table, I would propose to include data regarding compounds 2b - 6g in one line due to the identity of the results presented for them.

- in my opinion, it is unjustified to refer three times in chapter "3. Discussion" to one's own works that have not yet been published, by using the phrase "paper in preparation".

After taking into account these very minor, suggested corrections, I recommend that the work be accepted for publication in IJMS without re-evaluation.

Author Response

Reviewer 2:

The work submitted to me for review, concerning the search for new antifungal substances for specific use in Candida infections, concerns extremely important and current topics. A very good direction of research in this field is to pay attention to the inhibition of the ergosterol synthesis and consequently reduction of ergosterol levels by HMGR inhibitors, and the use of simvastatin as a reference substance in provided determinations. Although the results obtained are not impressive in all the presented areas, they are nevertheless important for the further effective development of this type of drugs.

Conceptually, the work is good planned, and it was made and presented very well. The scope of the research conducted is appropriate to draw the presented conclusions. Even though the tested compounds are not new substances, or perhaps because of it, this work is more biological and biochemical in nature and should be assessed in this form and scope. However, from a chemical point of view, I have some doubts about the use of the term "alpha-asarone analogues". The tested compounds are quite distant and highly differentiated analogues of azarone, which is considered as a standard substance. It is probably better to use everywhere the phrase used in the caption to Fig.2. Among typical editorial errors, I would like to point out the following:

- line 108 should be "SDD" instead of "SSD".

- in Table 1, in its header I would suggest swapping the words "Compound" and "Strain" and at the same time reversing the position of the diagonal line between these words.

- in the same Table, I would propose to include data regarding compounds 2b - 6g in one line due to the identity of the results presented for them.

- in my opinion, it is unjustified to refer three times in chapter "3. Discussion" to one's own works that have not yet been published, by using the phrase "paper in preparation".

After taking into account these very minor, suggested corrections, I recommend that the work be accepted for publication in IJMS without re-evaluation.

Responses at reviewer 2:

  1. We appreciate the reviewer's suggestions and change the term "alpha asarone analogues" to the reviewer's recommended phrase " the series of test compounds 1ab, 2ac, 3ab, 4ac, 5ab, 6ag, and 7ab”

Now line 75, 289, 607

  1. We changed SSD for SDD, now line 106.

  1. We made the changes suggested in Table 1. Additionally, we added data on the compounds (physicochemical characteristics) in supplementary material (now Table S1).

In addition, we complement the results (lines 121-122), discussion (lines 299-300) and materials and methods (lines 442-447) sections to justify the data of the suggested compounds.

  1. We eliminated the expressions "article in preparation", now line 401, “by our group” now line 390 and “our group”, now line 395.
